# Prevalence of Polymorphism and Post-Training Expression of ACTN3 (R/X) and ACE (I/D) Genes in CrossFit Athletes

**DOI:** 10.3390/ijerph20054404

**Published:** 2023-03-01

**Authors:** Omar Peña-Vázquez, Liliana Aracely Enriquez-del Castillo, Susana Aideé González-Chávez, Jaime Güereca-Arvizuo, Ramon Candia Lujan, Claudia Esther Carrasco Legleu, Natanael Cervantes Hernández, César Pacheco-Tena

**Affiliations:** 1Faculty of Physical Culture Sciences, Autonomous University of Chihuahua, Campus II, Circuito Universitario S/N, Chihuahua 31125, Mexico; 2PABIOM Laboratory, Faculty of Medicine and Biomedical Sciences, Autonomous University of Chihuahua, Campus II, Circuito Universitario S/N, Chihuahua 31109, Mexico; 3Department of Health Sciences, Multidisciplinary Division of Ciudad Universitaria, Autonomous University of Cd. Juárez, Ciudad Juárez 32310, Mexico

**Keywords:** alpha-actinin-3 human, angiotensin-converting enzyme, genotype, homozygote, heterozygote, physiology of exercise, real-time polymerase chain reaction, molecular biology, physical fitness, blood gene expression

## Abstract

Background: CrossFit is known as a functional fitness training high-intensity exercise to improve physical performance. The most studied polymorphisms are the ACTN3 R577X gene, known for speed, power, and strength, and ACE I/D, related to endurance and strength. The present investigation analyzed the effects of training on ACTN3 and ACE gene expression in CrossFit athletes for 12 weeks. Methods: the studies included 18 athletes from the Rx category, where ACTN3 (RR, RX, XX) and ACE (II, ID, DD) characterization of genotypes and tests of maximum strength (NSCA), power (T-Force), and aerobic endurance (Course Navette) were performed. The technique used was the reverse transcription-quantitative PCR real-time polymerase chain reaction (RT-qPCR) for the relative expression analysis. Results: the relative quantification (RQ) values for the ACTN3 gene increased their levels 2.3 times (*p* = 0.035), and for ACE, they increased 3.0 times (*p* = 0.049). Conclusions: there is an overexpression of the ACTN3 and ACE genes due to the effect of training for 12 weeks. Additionally, the correlation of the expression of the ACTN3 (*p* = 0.040) and ACE (*p* = 0.030) genes with power was verified.

## 1. Introduction

CrossFit (CF) has awakened great interest in applied sports sciences because it involves a high physiological demand and multiple variations to develop physical abilities and improves sports performance not only specifically in CF but also in a wide range of individual and team sports, which use it as a training method to improve performance. This is why the genetic expression of ACTN3 and ACE in CF athletes was analyzed, because it can be considered base training to potentiate the specific physical capacities of any other sport. In this way, the results can be extrapolated to other sports disciplines [1]. These trends are present in the ACSM Fitness Trends American College of Sports Medicine list [2]; however, environmental and genetic factors are strongly related to the athlete’s response to training, including improving physical strength, power, and aerobic endurance capacities [3].

Over the past two decades, understanding the genetic influence on sports performance has been a topic of great research interest, leading to identifying genes whose variants can help differentiate a potential elite athlete. The most studied polymorphisms are the R577X of the ACTN3 gene, commonly known as the gene for speed, power, and strength [4], and the I/D of the ACE gene, associated with the physiological traits of endurance and strength [5,6].

The ACTN3 gene encodes the synthesis of the protein α-actinin-3 (αA3) in fast twitch fibers [4]. Mutations in this gene result in the R and X alleles, which make it possible to synthesize, or not, the αA3 protein. The combination of alleles result in the RX, RR, and XX genotypes [7]. Individuals expressing XX are characterized by a total lack of αA3, associated with performance in highly demanding endurance sports, and it is estimated less than 20% of the world population possess it [8,9]. For their part, RR individuals, which exhibit a higher incidence in African Americans, express the αA3 protein in their fast muscle fibers, which are associated with maximum strength (MS) and speed performance [10,11,12,13,14].

The ACE gene encodes angiotensin-converting enzyme-1 (ACE-1) and contains the I and D alleles that give rise to the DD, ID, and II genotypes. These genotypes are a crucial element of the renin-angiotensin-1 system, responsible for blood pressure homeostasis and whose participation is fundamental in cardiorespiratory efficiency [6,15,16]. The DD genotype is associated with a higher percentage of type II (anaerobic) muscle fibers, providing an advantage in performing explosive type efforts. In comparison, the II genotype is associated with a higher percentage of type I (aerobic) muscle fibers [17], which predominate in sports that require high aerobic resistance [18,19,20].

In addition to the genotype, an essential factor is the involved genes and their expression levels, which increase as assumed during the adaptation to exercise and sports performance improvement due to training. Due to the effect of physical exercise, the existing literature on the expression of the ACTN3 and ACE genes has been carried out mainly in animals [13,21,22] and scarce studies have been conducted in humans [23,24,25]. To date, the effect of CF training on the changes in the expression of these genes, linking them with a direct impact on the improvement in sports performance, is not apparent.

According to the principle of functional unity, the fact that the human body is a single functional unit that works with all the organs and systems, which are interrelated with the others to the point that if one of these is left to work, continuity in training is impossible [26]), the study of a molecular mechanism provides information to help in the area of sports performance and therapeutics as a treatment for muscle injuries. Because the study methods are mostly invasive, it will be easier to identify the ACTN3 genotype and, thus, specifically address the injury recovery and new methodologies of physical exercise in patients with cardiovascular risk related to angiotensin-converting enzyme (ACE). Based on this, the hypothesis is that CF training increases the expression of ACTN3 and ACE, and the objective is to analyze the expression changes in the ACTN3 and ACE genes due to the training effect in CF athletes, with the specific aims of characterizing a sample of CF athletes from the RX category and determining their ACTN3 (RR, RX, XX) and ACE (DD, II, ID) genotypes, assessing physical fitness, determining the expression of ACTN3 and ACE genes pre- and post-training, and applying and assessing the volume of the training program.

## 2. Materials and Methods

Design: it is a quasi-experimental longitudinal correlational study; the sample was calculated using the sample size calculation equation for the comparison of two repeated paired means in a single group based on similar populations [27]. Change for: Finally, 18 CF athletes (7 women and 11 men with an average age of 27.7 ± 5.0 years, 16.0 ± 6.5 sports years, and 6.5 ± 2.6 years of CF training) were selected from the Rx category who had the level of training, competed internationally, and met the fastest execution times in the tests belonging to this category. They also belonged to the KingFit gym where they prepared and received their certified trainer level 1 and 2 from CrossFit Inc. (San Jose, CA, USA). Inclusion criteria: Rx category athletes between the ages of 18 and 35 and who did not have cardiac pathologies or musculoskeletal injuries that prevented them from carrying out training and evaluations. In the training macrocycle, the initial tests took place in the preparatory period of the athletes at the beginning of the year; after a 12-week training, the athletes repeated the same tests but they were already in the competitive period of the macrocycle, which corresponds because at that moment they had acquired their best sporting form of the season. We evaluated the physical capacities such as MS, power, maximum oxygen consumption (VO_2max_), and body composition (lean and fat mass).

Finally, the expression of the ACTN3 and ACE genes in blood and their genotypes were identified. Before applying any tests, the participants received an explanation, as well as the procedures and risks of the study. Subsequently, the athletes signed the informed consent based on the general health law on research and the Declaration of Helsinki. The Ethics Committee of the Faculty of Medicine and Biomedical Sciences of the Autonomous University of Chihuahua, under number CI-015-21, approved the study and its procedures.

### 2.1. Maximum Strength Test

The personal record (PR) in Front Squat (FSQ) and Shoulder Press (SHP) was used through the NSCA protocol (Spain) to assess the maximum force. First, a standardized, progressive, and specific warm-up protocol was carried out, starting with 5 min of activity on a stationary bike at 40–50% of the athlete’s maximum heart rate (HR_max_), followed by joint mobility exercises based on dynamic stretching with an emphasis on the spinal joint, hip, shoulders, lower and upper extremities. Later, it began with a period of familiarization with the exercise.

Finally, once the estimated PR for each athlete was determined, the loads were adjusted and progressive weight increases were made until reaching failure, thus, determining their PR, which was recorded in pounds (lbs).

### 2.2. Power Test

The test was carried out with the T-Force System encoder model TF-100 (Spain) for lower and upper limbs in conjunction with the Power Clean (PC), and Split Jerk (SJ) exercises with an Olympic bar of 45 lbs for men and 35 lbs for women. The maximum weight of the mentioned exercises was used for each athlete to delimit 35, 55, and 75%. Subsequently, the athletes performed three repetitions with each percentage of the PC and SJ exercises registered in newton.

### 2.3. Strength Assessment of the Anteflexion Muscles of the Arm

Digital Hand Dynamometers, Smedley III (Japan), were used to ensure a general strength index of the flexor muscles of the right and left hand; the test was performed in duplicate with a rest period between each limb, recording the highest value in kilograms (kg).

### 2.4. VO_2max_ Assessment

The estimation of VO_2max_ was determined with the Course Navette test [28], where the subject had to run 20 m back and forth, previously delimited and marked visually to the audio sound. Registration of the participant completing as many reps as possible was saved, and it was calculated according to the Ramsbottom formula [29]. The maximum level determined by the test was line 16 of level 2 with a total time of 22.19 min equivalent to 84.8 mL/kg/min for their VO_2max_, making sure to reach maximum intensities (>90% FCM) and supervised with the POLAR RS300 cardiac monitor (Polar Electro, Bethpage, NY, USA).

### 2.5. Body Composition Assessment

Following the International Standards for Anthropometric Measurement Manual, the testing of variables such as weight and height was under the guidelines of the International Society for the Development of Kinanthropometry (ISAK). Subsequently, with the data obtained, the body mass index (kg/m^2^), lean mass (kg), and fat mass (kg and %) were determined through the digital electrical bioimpedance equipment InBody-230 Biospace Co. (Seoul, Republic of Korea). Before the evaluation to run tests, they were asked to come fasting, with the minimum clothing for the test, empty their bladders, and refrain from exercising or alcohol consumption the day before the test.

### 2.6. RNA and DNA Extraction

A blood sample was obtained from the median cubital vein for a genetic test. RNA extraction was obtained by using the Direct-zol™ RNA Miniprep Plus TRIzol^®^ In kit (Zymo Research, Irvine, CA, USA), and the E.Z.N.A.^®^ MicroElute Genomic DNA Kit (Omega Bio-Tek, Norcross, GA, USA) was used for DNA extraction by following the manufacturer’s protocols.

### 2.7. ACTN3 Genotype

A blood sample was taken from the median cubital vein for genetic tests. To carry out the DNA extraction, we used the Direct-zol™ RNA Miniprep Plus TRIzol^®^ In kit (Zymo Research, USA) and the E.Z.N.A.^®^ MicroElute Genomic DNA Kit (Omega Bio-Tek, USA) for DNA extraction, according to the manufacturer’s protocols.

The polymerase chain reaction (PCR) was used to amplify the 291 base pair fragments of the ACTN3 gene using the primers FW: 5′-CTGTTGCCTGTGGTAAGTGGG-3′ and RV: 5′-TGGTCACAGTATGCAGGAGGG-3′ and the Radiant 2X TaqMastermix kit (Radiant Molecular Tools, South San Francisco, CA, USA). A final volume of 50 µL was used, with a final primer concentration of 240 nM and 2 µL of DNA. PCR was performed in a 2720 Thermal Cycler (Applied Biosystems™, Bedford, MA, USA) using the following amplification parameters: initial denaturation at 95 °C for 1 min, followed by 40 cycles of denaturation at 95 °C for 15 s, alignment at 55 °C for 15 s, and extension at 72 °C for 15 s.

By executing the restriction fragment length polymorphism (RFPL) technique, we could identify ACTN3 gene polymorphisms. First, the PCR products were quantified using a Nanodrop 200 spectrophotometer (Thermo Scientific, Waltham, MA, USA), then, 4 mg of each PCR product was digested with the enzyme *DdeI* (“*Desulfovibrio desulfuricans*”) (BioLabs, Inc. Beverly, MA, USA) according to the provider’s instructions. The final volume of each digestion mixture was 20 μL, which included 2 μL of 10X enzyme buffer, 1 μL of *DdeI* enzyme at 10,000 U/mL, the corresponding volume of the PCR product, and molecular biology grade water to complete the 20 μL. Finally, each mix was incubated in a water bath for 4 h at 37 °C and then 20 min at 65 °C to stop the reaction.

Each digested product was analyzed in a vertical 10% polyacrylamide-bis-acrylamide gel electrophoresis under denaturing conditions using the Mini-PROTEAN Tetra Cell system (Bio-Rad, Hercules, CA, USA) for 2 h at 38 V and 1 h at 90 V. Molecular weight markers of 50 and 25 base pairs (bp) were used as reference. For visualization, ethidium bromide (Invitrogen, Life Technologies, Waltham, MA, USA) was used at a final concentration of 10 µg/µL and the Benchtop 3UV™ transilluminator (UVP, Upland, CA, USA). The RR genotype was defined when the digestion pattern resulted in two bands of 205 bp and 86 bp); the RX genotype showed four bands of 205 bp, 108 bp, 97 bp, and 86 bp; and the XX genotype showed three bands of 108 bp, 97 bp, and 86 bp.

### 2.8. ACE Genotype

For the amplification of intron 16 of the ACE gene, the PCR was conducted using the primers FW: 5′-CTGGAGACCACTCCCATCCTTTCT-3′ and RV: 5′-GATGTGGCCATCACATTCGTCAGA-3′ and the Radiant 2X TaqMastermix kit (Radiant Molecular Tools, USA). A final volume of 25 µL was used with a final primer concentration of 240 nM and 2 µL of DNA. PCR was performed in the 2720 Thermal Cycler (Applied Biosystems™, USA) using the following amplification parameters: initial denaturation at 95 °C for 5 min, followed by 35 cycles of denaturation at 95 °C for 30 s, alignment at 67 °C for 30 s, and extension at 72 °C for 30 s.

The PCR product was analyzed by horizontal electrophoresis on 2.5% agarose gel using a 50 bp molecular weight marker. The expected amplification was 190 bp for the DD genotype, 190 and 490 bp for the ID genotype, and 490 bp for genotype II. The images were analyzed on a Benchtop 3UV™ documenter (UVP, USA).

### 2.9. ACTN3 and ACE Expression

The expression of ACTN3 and ACE was conducted by reverse transcription-quantitative PCR (RT-qPCR) using the specific primers for ACTN3 (FW: 5′-TCAGTTCAAGGCAACACTGC-3′ and RV: 5′-CCCACTTGGTGTTGATGTCC-3′) and ACE (FW: 5′-CGCAGAGCTACAACTCCAGCGCC-3′ and RV: 5′-GCCCCAGGCCTCCGCAAACTC-3′), using the reference gene glyceraldehyde-3-phosphate dehydrogenase (GAPDH) (FW: 5′-TGTTGCCATCAATGACCCCTT-3′ and RV: 5′-CTCCACGACGTACTCAGCG-3′).

Complementary DNA (cDNA) was synthesized from total RNA (1 µg) by RT using the SupeScript^®^ III First-Strand Synthesis Kit (Invitrogen from Thermo, Fisher Scientific) with random hexamer primers in a final volume of 20 µL in the 2720 Thermal Cycler (Applied Biosystems, USA). After the RT reaction, the volume reached 100 µL with molecular biology-grade water.

The RT-qPCR was performed using the Radiant™ SYBER Green Hi-ROX RT-qPCR Kit (Radiant) in a volume final of 20 µL. Three µL of each cDNA was used in each reaction, and a final primer concentration of 600 nM for ACTN3 and ACE or 400 nM for GAPDH. The RT-qPCR was carried out in the Quant Studio 3 PCR system (ThermoFisher Scientific) with a program that included: 1 cycle at 95 °C for 2 min, 40 cycles of denaturation at 95 °C for 5 s, and alignment and extension at 60 °C (ACTN3 and GAPDH) or 55 °C (ACE) for 30 s. The capture of fluorescence data occurred in real time during the extension step of each cycle. Each cDNA sample was tested in triplicate. The execution of relative quantification (RQ) was performed using the ΔΔCt method (RQ = 2^−ΔΔCt^).

### 2.10. Static Analysis

Differences between pre- and post-training assessments were determined using Student’s *t*-test. The correlations between the variables were estimated with the Pearson correlation coefficient. Statistically significant differences were considered when *p* ≤ 0.050.

All analyses were performed using the statistical package SPSS V.21 for Windows.

## 3. Results

### 3.1. Results Associated with General Characteristics

Table 1 shows the general characteristics of the population, classified as men and women. The values for the evaluated variables are age expressed in years, weight in kilograms, height in meters, body mass index in kilograms over meters squared, fat in kilograms, percentage of body fat in percent, and lean mass in kilograms. High body mass index levels in both genders were identified due to their extensive training frequency according to a high level of lean mass and low levels of body fat.

### 3.2. Results Associated with Genotype Frequency in the Population

For the ACTN3 gene, the frequency of the RX genotype was the highest (50%), followed by RR (22%) and XX (28%) (Figure 1), while for the ACE gene, the most prevalent frequency was that of the ID genotype (61%), followed by DD (22%) and II (17%) (Figure 2).

### 3.3. Results Associated with the Effect of Training on Physical Abilities

Table 2 shows the women’s values during the test before and after the training. A statistically significant difference was observed in the physical capacities of MS in the right arm, the MS of FSQ, the MS of SHP, and power in PC at 35%.

The presentation of values in men is in Table 3. During the pre-and post-training tests, significant differences were found in the value of the physical capacities of FSQ MS, SHP MS, PC power at 55 and 75%, and SJ power at 55 and 75%.

### 3.4. Results Related to the Association of ACTN3 and ACE Genotypes with Physical Abilities

The ACTN3 RR genotype was correlated with the strength of the flexor muscles of the right arm and the power of medium and heavy weights. These are connected to the increase in most of the physical capacities evaluated.

On the other hand, the correlations with the ACTN3 RX genotype mainly occurred with the MS of the FSQ and power with heavy weights. Therefore, they are primarily related to the increased physical capacities evaluated. Furthermore, among the different genotypes evaluated, it was the only one related to the increase in the MS of the FSQ with the VO_2max_.

With the ACTN3 XX genotype, the correlations were mainly with the increase in the MS of the SHP and the power with low weights; the association was with the increase in most of the physical capacities evaluated.

The correlations with the ACE DD genotype were mainly with the increase in the strength of the flexor muscles of the right arm and the power with low and moderate weights; these are related to the increase in most of the physical capacities evaluated.

The observation of correlations with the ACE ID genotype, mainly with the increase in maximum FSQ strength and power with low weights, is related to the increase in most of the physical capacities evaluated.

In the case of the ACE II genotype, the correlation was mainly with the increase in the MS of the left arm flexor muscles, directly related to the increase in power with low weights. On the other hand, the increase in the MS of the SHP decreases the power with medium weights, and the increase in the power with medium weights decreases the power with high weights. Correlations between physical abilities by genotype can be seen in the Appendix A.

### 3.5. Results Associated with a Correlation of ACTN3 and ACE Expression and Physical Abilities

Table 4 indicates the values of the ACTN3 and ACE expression (RQ = 2^−ΔΔCt^) and the evaluated physical capacities. The *p*-value was significant in the physical power capacity in PC at 35% for both genes.

### 3.6. Results Associated with Gene Expression of the ACTN3 and ACE Genes

Figure 3 shows the RQ values of the ACTN3 and ACE genes expression before and after training using the GAPDH gene as a reference. The CF training increased the expression of ACTN3 and ACE by 3.0 and 2.3 times, respectively.

## 4. Discussion

This is the first time that an overexpression of the ACTN3 and ACE genes regardless of genotype has been reported in CF athletes after training from blood samples. An overexpression of ACTN3 and ACE genes in blood samples is reported regardless of the genotype in CF athletes after training. It is a relevant finding in the field of sports training and molecular biology due to the methodology used, as well as the results.

In the characterization of subjects with the ACTN3 gene, 22% of the RR genotype was identified with 50% RX and 28% the XX, which is outstanding data since this quantity was pointed out for the first time in Mexican athletes. The identification for characterization of the ACE gene resulted in 22% of the DD genotype, 61% of the ID, and 17% of II.

Peña et al. [30] identified 26% of the RR genotype in volleyball players and 13% in tennis players, 74% of the RX in volleyball players and 81% in tennis players, unlike Güereca et al. [31], who found 25% of the RR genotype, 50% for the RX, and 25% for the XX in the physically active Mexican population. For the test of the ACE gene, Peña et al. [30] identified 20% in tennis players and 12% in volleyball players for the DD genotype, 63% in tennis players and 61% in volleyball players for the ID, and 17% in tennis players and 27% in volleyball players for II. It should be noted that the characterizations mentioned are in the Mexican population, as in the present study, so the research provides data on a characterization of these genotypes in the same population.

Concerning physical capacities, an increase was observed in MS in the right arm, FSQ, and SHP, and power in PC at 35% of PR in women. The previously mentioned contrasts with a similar study in female weightlifting athletes, which did not show an increase in power but did show a correlation between power levels at 30% isometric force and mean testosterone levels [32]. In contrast, men had higher performance in FSQ and SHP maximal strength, as well as PC and SJ power at 55% and 75%, possibly due to maximal strength. Shephard [33] describes that an individual’s MS is approximately proportional to third of his height. Hence, a woman who is 0.1 m shorter than a man inevitably tends to have a disadvantage of 20%.

During the second test, there were no changes in body composition (weight, lean mass, or fat mass). This is because the structuring of the body composition was not part of the objective of the macrocycle since the training increases the muscular activation and the system’s adaptation, which, in athletes, may not modify the physical components [32,34].

The RR genotype relates to the power evaluated with moderate and high loads. It is similar to the characteristics of the DD genotype, which show a relation with the increase in power with low and moderate loads, hence, the favorable effect for speed/power performance in track and field, weightlifting, speed skating, track cycling, and short and medium distance swimmers from Israel, China, Poland, UK, Russia, Australia, Greece, Italy, Portugal, Spain, Taiwan, United States, Israel, Japan and Commonwealth of Nations [15,35,36,37,38,39,40] confirms these genotypes as specific for speed power and high competitive level [12,36,41].

The results obtained for the RX genotype are associated with the MS of FSQ and power with high weights, having a similarity with the characteristics of the ID genotype, which show a relation with the increase in maximum force in FSQ and power with low weights based on the ability to provide more excellent stability of the structure of the contractile elements of the muscle fiber. This justifies a greater capacity for force production [7] consistent with the performance in sports that share physical capabilities that determine power and aerobic/anaerobic resistance [37], as shown by authors such as Ahmetov and Fedotovskaya [15], Boraita et al. [42], Myerson et al. [19], Papadimitriou et al. [17], Tobina et al. [43], Wang et al. [40], and Znazen et al. [44], who confirm an association with the endurance/power in middle-distance sports athletes from Poland, the United Kingdom, Russia, Australia, Greece, Italy, Portugal, Spain, Taiwan, the United States, Israel, Japan, and the Commonwealth of Nations. In addition, the RX genotype was the only one that related the increase in MS with VO_2max_. Even though the terms around the specific influence of the RX genotype have not been unified, since authors such as Ahmad et al. [45] and a meta-analysis carried out by Ma et al. [6] do not find a specific relationship in the influences, as mentioned above, it is not possible to make specific conclusions. However, it is a starting point for new findings regarding the RX genotype.

The XX genotype had a relationship with the MS in the SHP and the power with low weights, similar to the genotype II characteristics. However, this had an inverse correlation with the physical capacities evaluated, which refers to increasing MS power decreases and vice versa. This has an advantage over activities that only high aerobic resistance influences [6,46] and the maintenance of strength levels by its type I fibers [4,46].

Since the alpha-actinin-3 protein is only expressed in fast-twitch fibers, as mentioned by Pickering and Kiely [4], said deficiency of the XX genotype can be regulated by alpha-actinin-2 [24]. By genotype II, the ACE gene inhibitor temocapril produces a recovery of the type I muscle fiber composition ratio [46]. Therefore, the increase in the MS and power of these genotypes in the present investigation could be focused on the development of aerobic resistance performance [47]. With these claims for both genotypes, an association was reported with the aerobic performance of Ironman lead climbers, runners, swimmers, and triathletes from Poland, Russia, Austria, the United Kingdom, and other unspecified world-event-participating nationalities [18,19,20,47].

Domanska et al. [23] reported an increase in the expression levels of the ACTN3 gene in athletes, similar to Norman et al. [24], but in non-athlete subjects with prior training. On the other hand, Valdivieso et al. [25] found overexpression of the ACE gene in untrained subjects; however, they also mention that high-intensity training partly cancels the genetic influences mediated by the expression of the ACE gene.

Hence, a fact due to the uncontrollable training of the subjects, a training plan aims to achieve the athlete’s optimal performance and physical condition at a specific moment of the competition. In the present investigation, the training was systematized so we can attribute that there was an overexpression of the ACE gene three times greater regardless of the genotype, even though authors such as Seip et al. [48] report in their observations that the gene expression can be found in acute periods that do not exceed 24 h.

In the athletes evaluated in this research, a previous rest period of 48 h before extraction was considered as inclusion criteria, thus, confirming the theory of Egan et al. [49], who mention that the expression levels of the accumulated training depend on each protein.

When analyzing physical capacities with the expression of ACTN3 and ACE, a correlation was found in power (*p* = 0.040; *p* = 0.030), respectively, as well as in the research by Domańska et al. [23], who also found a relation between the expression of ACTN3 with power in athletes, unlike Norman et al. [24], who did not observe an association. This is possible because their subjects were a physically active population and not athletes. Therefore, controlled training justifies the relationship between gene expression and evaluated physical abilities. Physiological adaptations to training should be sufficiently established to reveal possible correlations. Regarding the genotypes, the carriers of the RR genotype of the ACTN3 gene have similarities when related to the DD genotype of the ACE gene, with a predominance in explosive sports. For their part, the carriers of the RX genotype have similarities with the ID genotype, with a predominance in strength and speed resistance sports. Finally, genotype XX is related to genotype II with a predominance in sports with characteristics of aerobic resistance.

Limitations: The main investigations focus their results on muscle biopsies, and although this method would be adequate, it would be repaired to detect an overexpression in blood; in turn, there was no control group to compare gene expression. Another limitation of the study was that the frequency of the ACTN3 gene RR genotypes and the ACE gene ID was insufficient to characterize the population and relate it to an overexpression of the genes due to the effect of training in specific genotypes.

## 5. Conclusions

The effect of training on the expression of ACTN3 and ACE genes in blood was examined. Our results indicate an overexpression (pre vs. post training) of the genes indicated in athletes. Additionally, the correlation of the expression of the ACTN3 and ACE genes with the power at 35% of the maximum weight was verified.

This research provides a novel and less invasive methodology in the analysis of the expression of the ACTN3 and ACE genes, extrapolated to the clinical and therapeutic area, specifically on injury reduction issues, in addition to supporting the effectiveness of training and the selection of models based on molecular talent.

Outlook statement: From these results, solid evidence is provided about sports sensitivity determined by genotypes and demonstrated in the physical specialty. It provides a starting point for new lines of inter and transdisciplinary research from the field of selection of sports talent through biological maturation, while in the field of health, it addresses new lines of work on injury recovery and new physical exercise methodologies in patients with cardiovascular risk.

## Figures and Tables

**Figure 1 ijerph-20-04404-f001:**
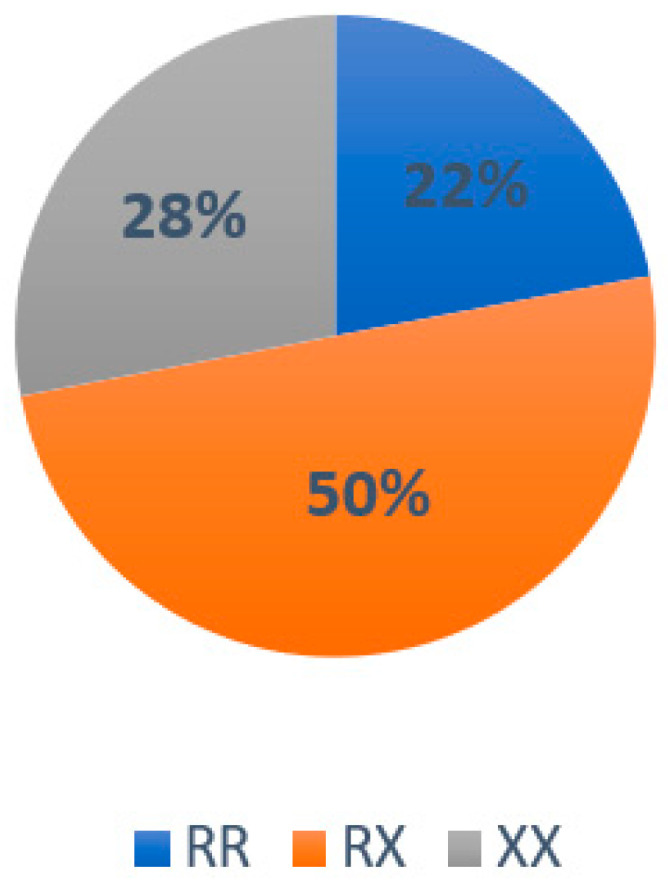
Genotype distribution of the ACTN3 gene in the study population.

**Figure 2 ijerph-20-04404-f002:**
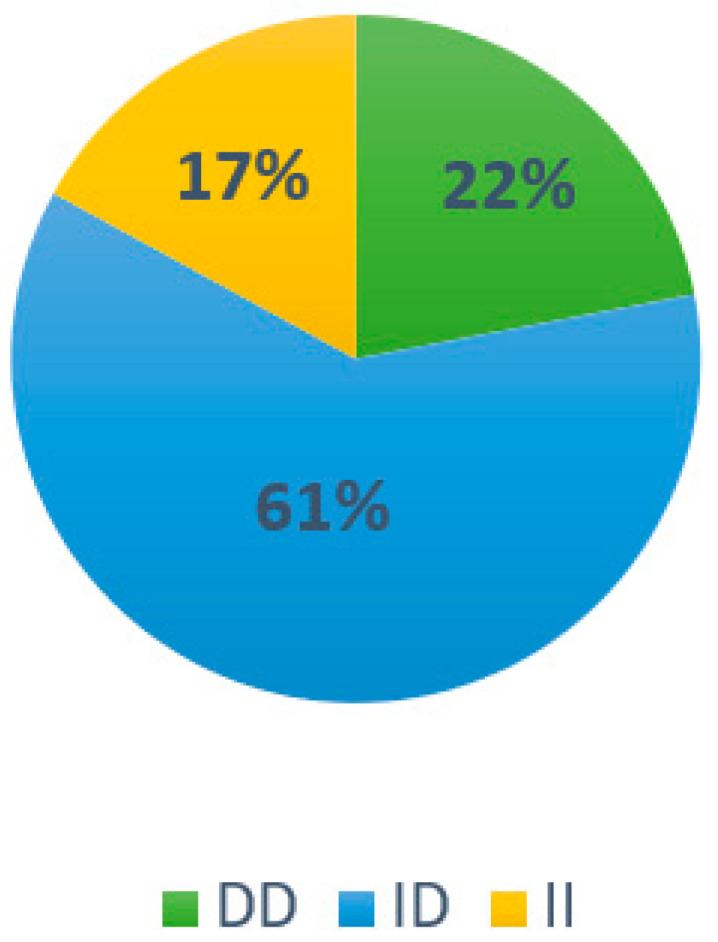
Genotype distribution of the ACE gene in the study population.

**Figure 3 ijerph-20-04404-f003:**
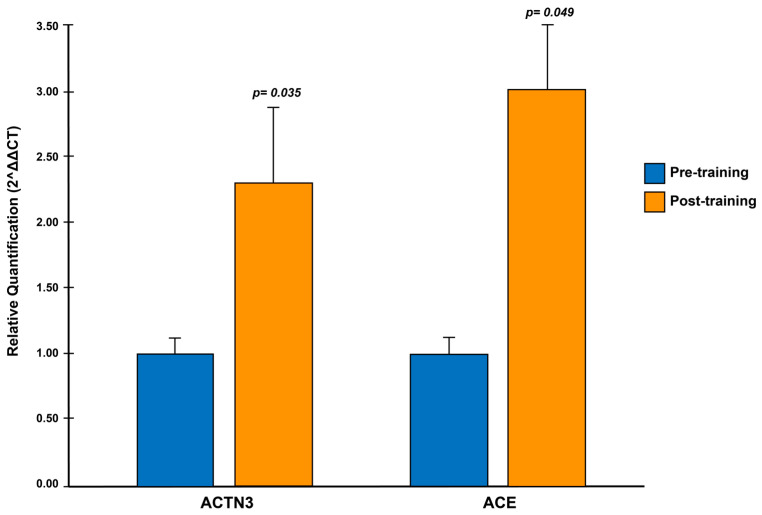
Relative quantification of the expression of the ACTN3 and ACE genes using the method 2^−ΔΔCt^.

**Table 1 ijerph-20-04404-t001:** Characteristics of CrossFit athletes.

Variable	Women		Men	
	Mean	SD	Mean	SD
Age (years)	27.1	±4.8	28.2	±5.6
Weight (kg)	60.5	±7.1	81.82	±11.3
Size (m)	1.6	±0.05	1.7	±0.06
Fat (kg)	13.3	±6.1	15.2	±7.0
PBF (%)	20.2	±4.1	18.3	±6.2
Lean mass (kg)	26.6	±2.5	38	±5.0

SD = standard deviation; Kg = kilograms; PBF = Percentage of body fat.

**Table 2 ijerph-20-04404-t002:** Physical capacities before and after training.

Variable	Women						
	Pre			Post			
	Mean		SD	Mean		SD	*p*
Left dynamo (N)	30.00	±	6.116	32.70	±	4.999	0.061
Right dynamo (N)	32.06	±	5.485	36.53	±	5.781	0.019 *
RM Front SQ (lbs)	166.87	±	32.175	176.87	±	30.698	0.010 *
RM Shoulder P (lbs)	90.00	±	7.559	95.00	±	10.351	0.050 *
VO_2max_ (mL/kg/min)	38.18	±	4.826	38.53	±	4.537	0.811
PPower35 (N)	1229.16	±	242.562	1307.5	±	299.983	0.025 *
PPower55 (N)	1676.00	±	446.957	1612.05	±	421.628	0.281
PPower75 (N)	1957.73	±	463.770	1895.91	±	391.960	0.484
PSplitj35 (N)	738.88	±	203.122	835.37	±	317.444	0.196
PSplitj55 (N)	1676.00	±	446.957	1612.05	±	421.628	0.281
PSplitj75 (N)	1957.73	±	463.770	1895.91	±	391.960	0.484

Left dynamo = Left dynamometry; Right dynamo = Right dynamometry; RM Front SQ = Front squat maximum repetition; RM Shoulder P = Maximum repetition of shoulder press; VO_2max_ = Maximum oxygen consumption; PPower35 = Power in power clean at 35%; PPower55 = Power in power clean at 55%; PPower75 = Power in power clean at 75%; PSplit35 = Power in split jerk at 35%; PSplit55 = Power in split jerk at 55%; PSplit75 = Power in split jerk at 75%; N = Newtons, * = *p* ≤ 0.05.

**Table 3 ijerph-20-04404-t003:** Physical capacities before and after training.

Variable	Men						
	Pre			Post			
	Mean		SD	Mean		SD	*p*
Left dynamo (N)	49.15	±	3.915	50.88	±	3.672	0.072
Right dynamo (N)	51.44	±	3.482	52.54	±	4.263	0.160
RM Front SQ (lbs)	287.00	±	35.683	301.75	±	31.314	0.007 *
RM Shoulder P (lbs)	159.50	±	14.424	165.00	±	14.529	0.003 *
VO_2max_ (mL/kg/min)	43.24	±	4.538	45.00	±	3.924	0.178
PPower35 (N)	2314.00	±	447.057	2085.85	±	364.059	0.051
PPower55 (N)	2905.73	±	433.930	2440.71	±	520.308	0.003 *
PPower75 (N)	3022.76	±	450.317	2544.44	±	485.093	0.010 *
PSplitj35 (N)	1317.68	±	385.735	1320.23	±	319.553	0.980
PSplitj55 (N)	2905.73	±	433.930	2440.71	±	520.308	0.003 *
PSplitj75 (N)	3022.76	±	450.317	2544.44	±	485.093	0.010 *

Left dynamo = Left dynamometry; Right dynamo = Right dynamometry; RM Front SQ = Front squat maximum repetition; RM Shoulder P = Maximum repetition of shoulder press; VO_2max_ = Maximum oxygen consumption; PPower35 = Power in power clean at 35%; PPower55 = Power in power clean at 55%; PPower75 = Power in power clean at 75%; PSplit35 = Power in split jerk at 35%; PSplit55 = Power in split jerk at 55%; PSplit75 = Power in split jerk at 75%; N = Newtons, * = *p* ≤ 0.05.

**Table 4 ijerph-20-04404-t004:** Results associated with a correlation of ACTN3 and ACE expression and physical abilities.

Variable	Correlation with ACTN3 Expression		Correlation with ACE Expression	
	Pearson	*p*	Pearson	*p*
Left dynamo (N)	0.344	0.162	0.400	0.100
Right dynamo (N)	0.252	0.312	0.278	0.264
RM Front SQ (lbs)	0.434	0.072	0.353	0.151
RM Shoulder P (lbs)	0.431	0.074	0.385	0.115
VO_2max_ (ml/kg/min)	0.041	0.871	−0.106	0.675
PPower35 (N)	0.489	0.040 *	0.512	0.030 *
PPower55 (N)	0.421	0.082	0.433	0.073
PPower75 (N)	0.412	0.089	0.425	0.079
PSplitj35 (N)	0.218	0.385	0.336	0.173
PSplitj55 (N)	0.376	0.124	0.407	0.094
PSplitj75 (N)	0.350	0.155	0.432	0.074

DS = Standard Ideviation; Kg = kilograms; PCG = Body fat percentage; N = Newtons; * = *p* ≤ 0.05.

## Data Availability

The data are not publicly available to maintain confidentiality. Data are available under justified request with the corresponding author, conditioned to the scientific committee approval and CONACyT with scholarship number 1063583.

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
