# Peer review of "Prevalence of Polymorphism and Post-Training Expression of ACTN3 (R/X) and ACE (I/D) Genes in CrossFit Athletes"

_ijerph, 2023, doi:10.3390/ijerph20054404_

Round 1

Reviewer 1 Report

The present investigation analyzed the effects of training on ACTN3 and ACE gene expression in CrossFit athletes for 12 weeks. Thank you for the opportunity to review your work, which is interesting but needs a little fine-tuning. My comments are set out below

Abstract: divide into sections Background, Methods, Results, Conclusion 

Line 35- add reference

Methods

Start by adding a subsection, it is useful to include a flow chart of the study . Generally this section reads quite hard, try to present it more clearly . Recruitment of participants, inclusion and exclusion criteria should be clearly described . At the end of the chapter, the methods of statistical analysis should be presented in a separate section. 

Results, 

Table 1 - I don't understand what the ± symbol means when you put mean and variance in separate columns 

Table 2,3,4- staystance significance should be indicated either by bold or * , and explained below the table (currently there is bold and *) 

 Discusion 

It is worthwhile to list the limitations in a separate subsection 

Conclusion 

Add practical implication

Author Response

We appreciate your comments. We have addressed each of them and improved the sufficient background including relevant references to the research, as well as the description of the methods and the conclusion as requested.

Reviewer 2 Report

Overall, I am pleased to review this paper and agree that it contains very interesting results. However, there are a few issues that need to be resolved before acceptance for publication can be considered. The research studies for the first time the polymorphism and post-training expression of ACTN3 and ACE genes in CrossFit (CF) athletes. So, this manuscript provides detailed insight in the molecular mechanisms of top athletic performance and significantly improves scientific knowledge on the emerging field of the new training concept. The manuscript idea, concept, and the obtained data are of good quality and permit interesting conclusions. However, the main concern is the presentation of the aim of this study. The authors should explain in more detail why CF athletes are being studied and clarify their aim statement. In addition, a passage on statistical analysis should be inserted. Due to a number of issues in the manuscript and a proposed additional analysis of the effects of individual gene polymorphism on the increase in gene expression, a major review before acceptance would be suggested.

TITLE

Line 3. The authors wrote “ACTN3 and ACE in CrossFit athletes”. Would the terminology ACTN3 and ACE gene be more accurate?

ABSTRACT

Line 16. CF is functional fitness training not high-intensity exercise, see [1].

Line 20. In my opinion, “from the Rx category” is not sufficiently informative. Rx is a CF specific term and may not be understood without further explanation, see other comment on the materials and methods section.

Line 20. Please correct the grammar of the sentence.

Line 26. What is meant by potency?

Line 27. Why is ACE Inhibitors referred to here?

INTRODUCTION

Line 32-35. One sentence in the introduction about CF is not a sufficient introduction to the topic here. CF involves constantly varied functional movements performed at high-intensity. Are more precise description may useful here. What elements are used in this type of training? What is the goal of the training program and how does it differ from other sports? (e.g., prepare athletes for the “unknown and unknowable” by reaching a broad, general and inclusive fitness vs. training in specific disciplines; non-specialization of the athletes and so on … ) As a result, why is the analysis of gene polymorphism and expression of particular interest in CF athletes? So, please add a few more introducing statements about CF.

Line 49. The authors wrote “black race”. Please correct the wording. People of color may an alternative term.

Line 63-64. Which hypothesis should be analyzed in this study? Based on the data in the literature, what do the authors expect? E.g., based on the high-intensity functional movements and the strength-endurance parts of the WOD (MetCon), is gene expression assumed to be increased by CF? (For that more details of CF is necessary to e explained to the readers, see previous comment) Please add a hypothesis statement.

Line 65. The authors wrote “principle of the functional unit”, what is here meant, Link unclear.

Line 66-67. The authors wrote “will provide factual information to help in the area of sports performance and the therapeutic, clinical, and preventive health areas”. Link unclear, why therapeutic, clinical, and preventive health areas are here referred to?

Line 68. The authors wrote “because of the proven effect in CrossFit …”. What proven effect? The authors should clarify their purpose of this research and add a specific aim statement.

MATERIALES AND METHODS

Line 72. The authors wrote “Rx category”. What are Rx category CF athletes. Why been this category selected? Rx only describe athletes perform WODs as described. This is a less detailed description of the performance capacity. Are there any other selection criteria, such as a specific ranking in the CF open (e.g., Top 100)? Are the athletes in the sample competitive athletes in CF at all?

Side note: To apply the term CF to research the study should incorporates and/or assesses the true nature of the CrossFit Inc. brand, including the gym where the exercise regime took place, official classes and/or exercises, and with a certified CrossFit trainer.

The authors should clarify doubts on this issue.

Line 73. The authors wrote “16.0 ± 6.5 sports years”. That sounds like general sports experience and not CF specific training or? A description of the level of CF experience is missing here.

Line 74. The authors wrote “corresponding from the preparatory period to the competitive period”. Link unclear, what is meant here?

Line 78. The authors wrote “are searched”. Gene polymorphism don´t be searched, rather it is determined/ identified  …

Line 85. What PR? The One repetition maximum (1-RM)?

Line 88. The authors wrote “40-50%of heart rate”. Should it be the maximal HR (HRmax)?

Line 97. How was the maximum weight identified?

Line 105. The authors referend to kg and previous (in line 93-93) to lbs. The manuscript should should report units in a consistent manner.

Line 109. As many rounds as possible with what time limit?

Line 122. Why this gene polymorphism and their expression can be analyzed in blood samples after training?

Line 128-131. This sentence may be redundant, see 2.6 RNA and DNA extraction.

Line 152. For the identification of the genotypes, are reference substances also used, as positive controls?

Line 172. The authors correctly refer to RT-q-PCR as reverse transcription-quantitative PCR, due to RNA was transcribed into DNA prior PCR analysis. Please correct this in a consistent manner throughout the manuscript, see Line 22 “real-time q-PCR”.

Section on the statistical analysis of the data is missing.

RESULTS

Table 2 and 3. It should be VO2max, instead of VO2max. What is meant by Previous and Later? The authors should use more on the scientific vocabulary, e.g., pre-and post CF training.

Line 221. How was the correlation measured? A summary of the correlations in tabular form would be helpful to follow the argumentation of the authors.

Line 245. How is the correlation analysis performed, with the quantitative results of the gene expression via RT-q-PCR? Here, the results are difficult for readers to comprehend.

Table 4. The authors wrote “Correlation with ACE expression expression”. One term expression is redundant.

Line 250-252. Is the increase in gene expression dependent on the individual gene polymorphism? The data were collected as part of this study, and it would be very interesting for readers to have this analysis presented additionally in the results section.

Figure 3. Figure caption is missing.

DISCUSSION

Line 256. The authors wrote “regardless of the genotype”. How can the authors derive this statement without presenting data of the impact on the polymorphism genotype?

Line 258-259. It´s actually the first-time data on the ACTN3 and ACE gene of CF athletes were presented. It would be advisable to specify this statement.

Line 261. What proportion?

Line 262. Why are Mexican athletes referenced here, is it not research on CF athletes?

Line 305. Correct the spelling of VO2max.

Line 331. How was the training systematized? According the official 'CrossFit® Level 1 Training Guide'? To discuss the influence of the systematic progression during CF training it is necessary to clarify the training intervention in the material in methods section.

Line 331-334. The purpose of the statement is not clear. Please clarify the assumption.

What are the limitations of this study? The authors should add a critical assessment of their results and conclusions. In this study no control group is involved, therefore some assumptions of the influence of CF training on gene expression had to be limited for example.

CONCLUSIONS

Line 352. The authors wrote “according to gender”. The authors don´t present data of the gene expression depending on gender.

Line 353-357. The authors should add references on their statements, based on their presented data or literature references.

A outlook statement is missing. What can be next steps based on the study results?

1.            Dominski, F.H.; Tibana, R.A.; Andrade, A. " Functional Fitness Training", CrossFit, HIMT, or HIFT: What is the preferable terminology? Frontiers in Sports and Active Living, 207.

Author Response

Dear reviewer

We appreciate your comments which were taken care of. We hope that with your recommendations, the article has improved in quality. We hope to meet the expectations to be considered for publication in the journal.

Kind regards

Round 2

Reviewer 2 Report

The authors have followed the suggestions and comments. The manuscript is now substantially improved.